# Immunological and Virological Responses in Patients with Monoinfection and Coinfection with Hepatitis B and C Viruses in the Brazilian Amazon

**DOI:** 10.3390/tropicalmed10060166

**Published:** 2025-06-13

**Authors:** Joseane R. Silva, Regiane M. A. Sampaio, Patrícia F. Nunes, Vanessa S. Guimarães, Camila Carla da Silva Costa, Evelen da Cruz Coelho, Micheline Vale de Souza, Luana Wanessa Cruz Almeida, Hellen T. Fuzii, Aldemir Branco Oliveira Filho, Luisa C. Martins

**Affiliations:** 1Laboratory of Clinical Pathology of Tropical Diseases, Federal University of Pará (UFPA), Tropical Medicine Center (NMT), Umarizal, Belem 66055-240, Pará, Brazil; regianearnund@yahoo.com.br (R.M.A.S.); dsnpatriciaferreira@gmail.com (P.F.N.); farmanessa@hotmail.com (V.S.G.); camila.carla.costa@hotmail.com (C.C.d.S.C.); evelencoelho@hotmail.com (E.d.C.C.); michelinevale@yahoo.com.br (M.V.d.S.); luana.bless15@gmail.com (L.W.C.A.); hellenfuzii@gmail.com (H.T.F.); 2Center for Biological and Health Sciences, University of the State of Pará (UEPA), Belem 66045-250, Pará, Brazil; 3Central Laboratory of Pará (LACEN), Belem 66645-001, Pará, Brazil; 4Institute for Coastal Studies, Federal University of Pará (UFPA), Bragança 68600-000, Pará, Brazil; olivfilho@ufpa.br

**Keywords:** coinfection, hepatitis B virus (HBV), hepatitis C virus (HCV)

## Abstract

Infections with the Hepatitis B (HBV) and Hepatitis C (HCV) viruses share some transmission routes, which is why coinfection with these viruses becomes common, especially in endemic areas. This study evaluated the immunological response profile, viral load, and liver damage in groups monoinfected with HBV or HCV and in those co-infected with HBV/HCV. The groups were composed of 22 patients monoinfected by HCV, 22 patients monoinfected by HBV, and 34 co-infected by HBV/HCV, according to serological markers and molecular biology tests. The study was carried out from December 2017 to October 2019. Virus detection employed enzyme immunoassay, Enzyme-Linked Immunosorbent Assay (ELISA), and real-time PCR, while liver function and fibrosis were assessed using biochemical tests and Fibroscan. To research the immunological profile, cytokines were quantified using the BIO-Plex Pro Human Cytokine. Comparing the groups, both mono- and co-infected patients exhibited a Th1 immune response profile. HCV monoinfection notably showed significantly elevated serum levels of INF-γ (*p* < 0.01) and TNF-α (*p* < 0.01). The viral load was significantly higher in the HCV monoinfected group when compared to the other groups. Regarding liver damage, patients with a high level of fibrosis (F4) presented significant levels of cytokines INF-γ (*p* < 0.001), IL-17 (*p* < 0.0001), and TNF-α (*p* < 0.0001).

## 1. Introduction

According to the latest WHO estimates for 2022, approximately 254 million individuals are affected by hepatitis B, while around 50 million are affected by hepatitis C. Although new infections showed a slight decline from 2019, the overall prevalence of viral hepatitis remains substantial. In 2022, there were 2.2 million new infections reported, a decrease from 2.5 million in 2019 [1]. In Brazil, viral hepatitis is the leading cause of liver transplants [2]. The northern region of the country is considered an area of silent prevalence or hidden prevalence of HBV and HCV infections due to flaws in the notification system, the low number of published studies, and the difficulty in accessing some areas; thus, coinfection (HBV/HCV) is not uncommon in this region [3].

Coinfection (HBV/HCV) is a rare condition among these viruses that occurs in 2 to 10% of anti-HCV positive individuals and in 3 to 20% of patients positive for HBsAg [3,4,5]. In the North Region of the country, Sampaio et al. found a prevalence of coinfection (HBV/HCV) of 9.88% [6].

In the literature, the severity of liver damage in patients with coinfection (HBV/HCV) is contradictory, as some studies have demonstrated an association of coinfection with increased liver damage and risk of developing cirrhosis and hepatocellular carcinoma, while others do not report this association [7,8,9,10]. A study carried out in the North Region by Sampaio et al., found no significant differences in the degree of fibrosis when comparing co-infected and monoinfected patients [6].

In vitro studies indicate that HBV and HCV replicate in the same hepatocyte without interference [7,10]. However, coinfection exhibits different virological and immunological profiles when compared to monoinfection, so coinfection may result in lower levels of viremia of one or both viruses compared to monoinfection [11,12]. According to Konstatinou and Deutsch (2015), HBV replication is normally suppressed by HCV in coinfection [4].

The immunological response plays an important role in the progression of chronic viral hepatitis to cirrhosis, as inflammation is a persistent immunological response [13,14]. Additionally, cytokines play an important role in the defense against viral infection, either indirectly, through determining the predominant pattern of the host response, or directly, through the inhibition of viral replication [15]. In infections caused by hepatitis B and C viruses, the imbalance in the production of pro-inflammatory (Th1) and anti-inflammatory (Th2) cytokines may play an important role in the course of the infection [16]. However, with regard to HBV/HCV coinfection, there are few studies concerning the immunological response, especially with regard to the scope of cytokines involved in the inflammatory response. A study carried out by Chen et al. indicated that some cytokines of the Th1 profile, such as interleukins IL-6 and IL-8 and tumor necrosis factor -α (TNF-α), decreased in coinfection when compared to HCV and HBV monoinfection [11].

The immunological and virological responses together provide a greater understanding of viral infection in coinfection [11]. In this sense, the objective of this study was to evaluate the inflammatory response, through cytokines, as well as the viral load levels displayed in coinfection and monoinfection, in addition to describing the impact of these infections on the liver fibrosis score.

## 2. Materials and Methods

### 2.1. Study Population and Sample Collection

This cross-sectional study included 78 (*n* = 78) patients diagnosed with viral hepatitis B and/or C, recruited from the Viral Hepatitis Service of the Tropical Medicine Center at the Federal University of Pará, a reference unit for these infections in the city of Belém, Pará, Brazil. Participants were referred from public screening centers for sexually transmitted infections (STIs) and, after registration in the specialized service, underwent a complete medical evaluation followed by referral to the Clinical Pathology Laboratory for collection of biological samples and diagnostic testing. Participant selection was based on positive ELISA results for anti-HCV antibodies and/or hepatitis B surface antigen (HBsAg). By default, all individuals were also tested for anti-HBc antibodies. All individuals included in the analysis were males or females aged 18 years or older, thus constituting a representative sample of the adult population.

Patients who had comorbidities such as diabetes, obesity, and hypertension, as well as those coinfected with human immunodeficiency virus types 1 and 2 (HIV1/2), human T-cell lymphotropic virus types 1 and 2 (HTLV-1/2), or hepatitis D virus (HDV), and individuals with pro-inflammatory conditions, including gout and rheumatoid arthritis, were excluded from the study. Additionally, patients who were users of illicit drugs and those who had previously undergone antiviral treatment for any of the pathologies studied were excluded.

A 10 mL blood sample was collected from each patient by intravenous puncture using a vacuum collection system with ethylenediaminetetraacetic acid (EDTA) as an anticoagulant, then the samples were processed for serum separation and divided into three aliquots, one for cytokine levels, one for viral load quantification, and another for serology. All participants provided written informed consent prior to enrollment in the study, in accordance with the principles outlined in the Declaration of Helsinki. The study was approved by the Ethics Committee for Research Involving Human Beings of the Tropical Medicine Center of the Federal University of Pará, under protocol number 2.432.635.

### 2.2. Laboratory Tests

Biochemical analyses: To measure the liver enzymes AST/TGO and ALT/TGP, a methodology standardized by the manufacturer was used with the commercial kits InterKit TGO and InterKit TGP (Katal Biotecnológica, Belo Horizonte, MG, Brazil). The analysis of the samples was performed by spectrophotometry on Thermoplate Analyzer Prietest Touch semi-automated equipment (Katal Biotecnológica, Belo Horizonte, MG, Brazil).

Serological tests: HBV surface antigen (HBsAg) and specific anti-antibody for HCV (Anti-HCV) were detected using a qualitative immunoenzymatic ELISA Kit (Enzyme Linked Immuno Sorbent Assay) using Kit DiaSorin (Vercelli, Saluggia, Italy), and Kit DiaPro(Anti-HCV 4 generations) (Sesto San Giovanni, Milan, Italy). Tests were carried out according to the manufacturer’s instructions.

Molecular biology: For detection and quantification of HBV by real-time PCR and HCV by real-time RT PCR, the Abbott m24sp/m2000rt automated system was used with the Abbott Real Time HBV and HCV Kit (Abbott Laboratories, Chicago, IL, USA), which presents linearity from 10 IU/mL to 1,000,000,000 IU/mL and 12 IU/mL to 100,000,000 IU/mL, respectively.

Genotyping: HCV was genotyped by the Restriction Fragment Length Polymorphism (RFLP) technique using the AVAII and RSAI restriction enzymes. Positive and negative controls were included in all reactions [17].

### 2.3. Assessment of Liver Elasticity (Fibroscan Elastography)

Elastography was performed as described by Bonnard et al. All measurements were obtained from the right lobe of the liver through the costal space. Image acquisition was guided by ultrasound, and the area was measured with a depth ranging from 25 to 45 mm; 10 valid measurements were obtained per patient. The results were expressed in kilopascals (kPa) [18]. Liver stiffness corresponds to the average value of all valid measurements. Metavir fibrosis stages were classified based on kPa values as follows: F0 when 2.0 to 4.5 kPa, F1 when 4.5 to 5.7 kPa, F3 when 5.7 to 12, 0 kPa and F4 when 12.1 to 21.0 kPa.

### 2.4. Serum Cytokine Levels

Cytokine quantification was performed using the Bio-Plex Pro™ Human Cytokine 17-plex Assay kit (IL-1β, IL-2, IL-4, IL-5, IL-6, IL-7, IL-8, IL-10, IL-12, IL-13, IL-17, TNF-α, IFN-γ, MCP-1, MIP-1β, G-CSF, GM-CSF) (Bio Rad, Hercules, CA, USA), following the manufacturer’s recommendations. Samples were analyzed on the Luminex 100™ microsphere analyzer (Luminex^®^, MiraiBio, Alameda, CA, USA). The levels of cytokines were expressed in pg/mL.

### 2.5. Statistical Analysis

The collected data were entered into the database in Microsoft Office Excel 2013. To verify the normal distribution of cytokines and viral load, the Shapiro–Wilk test was used, and for the difference between epidemiological and prevalence information, Chi Square and ANOVA tests were used, adopting a value of *p* < 0.05. The software used for these statistical analyses was Bioestat, version 5.3. To compare differences between groups, the non-parametric Mann–Whitney and Kruskal–Wallis tests were used for independent samples, considering a significance level of 5% (*p* ≤ 0.05). GraphPad Prism 5.0 software was used to perform these statistical analyses and generate graphs.

## 3. Results

The samples were classified into three groups (HCV, HBV, HBV/HCV) according to the results of immunological and molecular biology tests. We selected 22 (*n* = 22) patients monoinfected by HBV (HBsAg (+), HBV DNA (+), HCV RNA (−), and Anti-HCV (−)), 22 (*n* = 22) patients monoinfected by HCV (Anti-HCV (+), HCV RNA (+), HBsAg (−), and HBV DNA (−)), and 34 (*n* = 34) patients co-infected with HBV and HCV (HCV RNA (+) and HBsAg (+) and/or HBV DNA (+)). Among the patients studied, the age range ranged from 26 to 74 years, the average age of the co-infected group was 47.3 years, that of the HBV monoinfected group was 42.1 years, and that of the HCV mono-infected group was 59.2 years. Comparison between groups showed that the majority of participants were male (Table 1).

The viral genotype for HCV was determined in the HCV co-infected and monoinfected groups, with genotype 1 being observed as predominant in both groups. The viral replication of HBV and HCV was evaluated by quantifying the plasma viral load in the monoinfected and co-infected groups; thus, it was observed that the viral load in the HBV monoinfected group was higher when compared to the co-infected group (*p* = 0.0001), as well as being higher in the HCV monoinfected group when compared to the co-infected group (*p* = 0.0005). Liver enzymes AST and ALT were higher in the HCV co-infected and monoinfected groups (*p* < 0.0001). To verify the progression of the disease, liver fibrosis was evaluated between the HCV co-infected and monoinfected groups. In this study, it was observed that severe fibrosis (F3/F4) was more frequent in the monoinfected group (Table 1).

Figure 1 and Figure 2 show the median cytokine levels of patients in the co-infected (HBV/HCV) and monoinfected (HBV and HCV) groups. All cytokines evaluated showed higher levels in the HCV monoinfected group.

Comparing the HCV monoinfected group with the co-infected group (HBV/HCV), among the cytokines that presented significantly higher serum levels, the following stood out: IFN-γ (*p* < 0.01), IL-2 (*p* < 0.05), IL-4 (*p* < 0.01), IL-5 (*p* < 0.05), IL-6 (*p* < 0.05), IL-12 (*p* < 0.01), and IL-13 (*p* < 0.001). Comparing the HCV monoinfected group with the HBV monoinfected group, the cytokines that were significantly elevated were: IFN-γ (*p* < 0.001), IL-4 (*p* < 0.001), IL-5 (*p* < 0.001), IL-6 (*p* < 0.05), IL-8 (*p* < 0.05), IL-10 (*p* < 0.01), IL-12 (*p* < 0.01), IL-13 (*p* < 0.0001), and TNF -α (*p* < 0.01).

Figure 3 and Figure 4 show the cytokines evaluated according to the liver fibrosis score in the HCV co-infected and monoinfected groups, respectively. All cytokines were significantly higher in the severe fibrosis score (F3/F4) in the HCV monoinfected group, with the exception of IL-5 (*p* = 0.0561), IFN-γ (*p* < 0.0019), IL -2 (*p* < 0.001), IL-4 (*p* < 0.0001), IL-6 (*p* < 0.0001), IL-8 (*p* < 0.0045), IL-10 (*p* < 0.0021), IL-12 (*p* < 0.0001), IL-13 (*p* < 0.0001), IL-17 (*p* < 0.0001), and TNF-α (*p* < 0.0001). In the co-infected group, there was no statistically significant difference between fibrosis scores.

## 4. Discussion

Some studies have shown that male gender and age over 40 years are predominant for individuals with HBV, HCV, and HBV/HCV infection, including a higher risk of coinfection (HBV/HCV) for males [6,10,19,20,21].

Hepatitis B and C are often silent infections, mentioned by their late clinical presentation, indicating that the average age for diagnosis of these infections is higher. In Brazil, it is worth noting that patients with hepatitis C tend to be more advanced in age compared to those with hepatitis B, possibly due to the prolonged asymptomatic nature of hepatitis C. In addition, studies indicate that hepatitis C has a considerably higher probability of becoming chronic when compared to hepatitis B, in immunocompetent individuals. According to the WHO, it is estimated that approximately 55% to 85% of people who contract HCV develop a chronic infection [1]. In Brazil, data from the Ministry of Health report that these rates reach 90% [22].

In the context of coinfection, the dominance of HCV over HBV is commonly described in the literature [4,23,24,25]. A study showed that the presence of HCV significantly inhibited HBV replication both in vitro and in vivo, whereas HBV did not affect HCV replication. This inhibition could be associated with the low levels of HBsAg or the increased production of the cytokine IL-10 in response to HCV infection [25,26].

The present study demonstrated that viral load was higher in the monoinfected groups compared to the co-infected group. As for the co-infected group, it is noted that HCV replicates more intensely than HBV, suggesting that in coinfection, there may be competition between viruses, with HCV predominating. Additionally, the presence of one virus appears to decrease the replication of the other, which is reflected in a lower quantification of the viral load of HBV and HCV in the co-infected group when compared to the HBV monoinfected and HCV monoinfected groups. In the North Region of the country, there is a higher prevalence of infection with the hepatitis C virus; therefore, it is possible that HCV is the first virus to infect a hepatocyte, which reflects a greater number of cells infected by this virus, resulting in higher serum viral loads [6].

Regarding the inflammatory response, this study showed that in the group monoinfected by HCV, most of the cytokines studied were higher in relation to the other groups, but with a predominance of pro-inflammatory cytokines mediated by the cytokines INF-γ and TNF-α, which suggests a Th1 inflammatory response profile. Shih et al. in their study, also highlighted that INF-γ showed a higher serum level in the HCV monoinfected group in relation to healthy controls [27]. Additionally, there is evidence that patients with HCV infection present a Th1 type response with overproduction of INF- γ [28,29,30,31].

The co-infected group showed a higher pro-inflammatory cytokine profile compared to the anti-inflammatory profile cytokines, demonstrating a possible tendency towards a Th1 inflammatory response profile. The group monoinfected by HBV demonstrated the lowest levels of cytokines detected in relation to the other two groups, highlighting a greater amount of Th1 cytokines, mediated mainly by TNF-α. Additionally, some research highlighted that patients with chronic hepatitis B had a Th1 response with higher levels of IFN-γ, TNF-α, and IL-2, when compared to patients with hepatitis C and healthy individuals [32,33].

In the study, the elevated levels of pro-inflammatory cytokines—especially in the HCV monoinfected group—may reflect a persistent immune response induced by viral inflammation, since both viruses activate inflammatory pathways, which can amplify this response. Although anti-inflammatory cytokines are also produced, the pro-inflammatory response predominates.

The lower levels of cytokines in the HBV monoinfected group may be attributed to its immunologically "silent" nature. Compared to HCV, HBV induces a less robust proinflammatory response, which not only explains the reduced cytokine levels but may also hinder its early detection by the immune system, favoring the establishment of a chronic and often asymptomatic infection.

In coinfection, as the presence of two viruses seems to reduce their replication but with a predominance of HCV, there is a lower production of cytokines in response to infection, when compared to the group monoinfected by HCV.

Liver enzymes alanine aminotransferase (ALT) and aspartate aminotransferase (AST) are important markers in the evaluation of liver diseases. ALT shows greater specificity for liver injury than AST due to its predominant localization in the cytosol of hepatocytes, in contrast to the multisystemic distribution of AST. In our study, the HCV-monoinfected group exhibited significantly higher ALT levels [33]. Although AST levels in this group were lower than in the coinfected group, they remained above the reference values. As a marker of fibrosis, ALT demonstrates high sensitivity (87–89%) but low specificity (34–35%) [34]. Despite its limited specificity, its high sensitivity, combined with its more restricted tissue distribution, makes it a valuable tool for the screening and monitoring of liver diseases, even though its serum levels may be influenced by factors unrelated to hepatic necrosis.

In this study, significantly elevated levels of Th1 cytokines were found in patients with severe fibrosis scores in the HCV monoinfected group. According to Falasca, IFN-γ and IL-4 are important in the chronic progression of hepatitis C [35]. Souza-Cruz et al. reported in their research higher serum levels of the cytokines TNF-α, IFN-γ, IL-2, IL-6, IL-4, and IL-17 in HCV patients with an F3/F4 fibrosis score, which could support these cytokines as biomarkers associated with this fibrosis score [36]. IL-17 is a chemoattractant cytokine for neutrophils and has been reported to be involved in many immunological processes, mainly in the induction and mediation of pro-inflammatory responses. Furthermore, studies have shown that IL-17 is higher in serum from patients with chronic hepatitis C [37,38]. Corroborating data found in this study, this cytokine has already been correlated with the severity of liver inflammation [39,40].

Although the present study showed higher levels of pro-inflammatory cytokines in the HCV monoinfected group, mediated mainly by IFN-γ, on the other hand, low levels of the cytokines IL-10 and IL-4 were demonstrated, which may highlight an attempt by the immune system in modulating excessive pro-inflammatory responses. Additionally, Souza-Cruz et al. found in their research low levels of IL-10 in patients with a fibrosis score F3/F4 in relation to those with a score ≤ F2, suggesting that the lower the production of this cytokine, the greater the inflammatory response and consequently excessive damage to liver tissue [36,41].

In the present study, levels of pro-inflammatory cytokines may be related to the severe fibrosis score as a reflection of the immune system’s attempt to control the infection, leading to an exacerbated response that may lead to severe fibrosis in addition to chronic liver damage. This mediation appears to be carried out mainly by IFN-γ, as it is a cytokine whose prolonged production can contribute to inflammatory damage. Lack of anti-inflammatory regulation can also contribute to chronic inflammation and liver damage.

These results are important because, considering the endemicity of these diseases, there are no studies in the Brazilian Amazon population that have identified the difference in cytokine profiles related to monoinfection and coinfection by HBV/HCV, integrating viral load data and liver markers. Additionally, in the literature, few studies concerned with immunological profiles are supported by a very varied panel of cytokines, which offers a more robust analysis of the immune response.

The study has some limitations. The sample size was small due to the difficulty in recruiting the target population, which requires caution in interpreting some findings. The observed age difference between the HBV and HCV groups reflects distinct epidemiological patterns. In addition, the lack of fibrosis data for the HBV monoinfected group, due to its non-inclusion in the public health service protocol, prevents a complete comparative analysis of disease progression, which could imply important data for therapeutic guidelines. Although data on alcohol consumption and genetic factors were not included in this study, this does not appear to have impacted the main results, since the prevalence of fibrosis did not differ significantly between groups. We recommend that future studies include comprehensive data on these variables, for more specific contributions.

## 5. Conclusions

These results suggest that the proinflammatory environment sustained by IFN-γ and TNF-α appears to accelerate fibrogenesis, and that in coinfection, despite showing active HCV replication, fibrotic progression was lower, which may reflect a complex viral interaction aligned with recently described molecular mechanisms, in which HBV protein X (HBx) stimulates the stabilization of HCV core proteins [42]. Thus, in this clinical context, it is necessary to prioritize prognostic parameters, such as the degree of fibrosis, in monoinfected and co-infected patients. In addition, the investigation of IFN-γ and TNF-α as candidate biomarkers seems to be of interest. Future studies should explore these intrahepatic cytokine networks in order to elucidate how HBV modulates the response to HCV at the tissue level.

## Figures and Tables

**Figure 1 tropicalmed-10-00166-f001:**
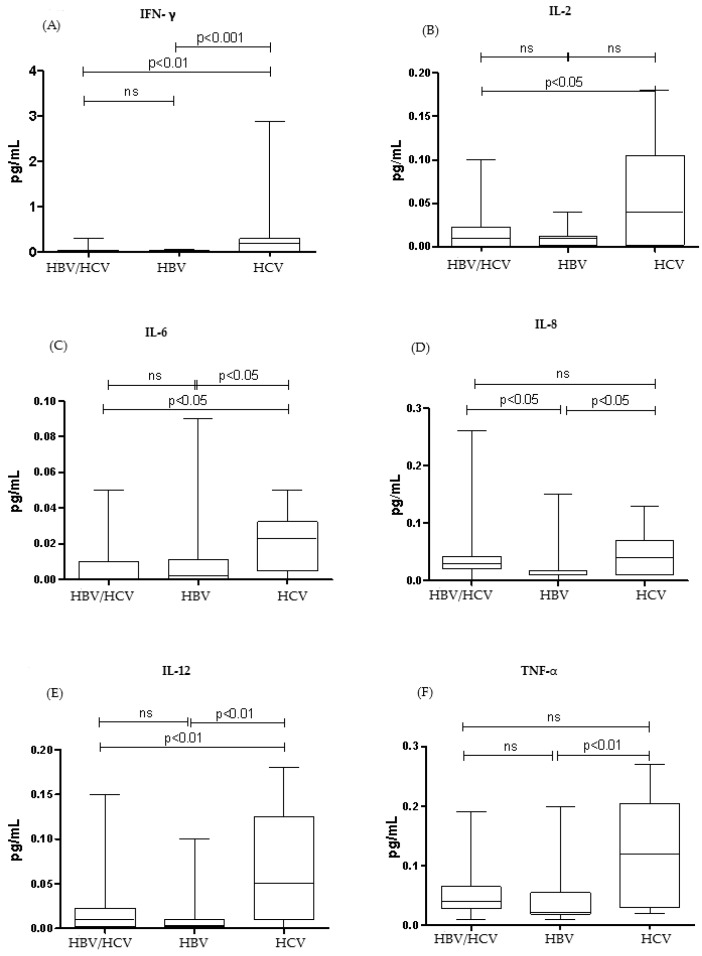
Th1 (pro-inflammatory) cytokine profile. Comparison of IFN- γ (**A**), IL-2 (**B**), IL-6 (**C**), IL-8 (**D**), IL-12 (**E**), and TNF-α (**F**) cytokine levels between co-infected (HBV/HCV), monoinfected (HBV), and monoinfected (HCV) groups. Data are expressed as median (25th–75th percentile). Kruskal–Wallis test.

**Figure 2 tropicalmed-10-00166-f002:**
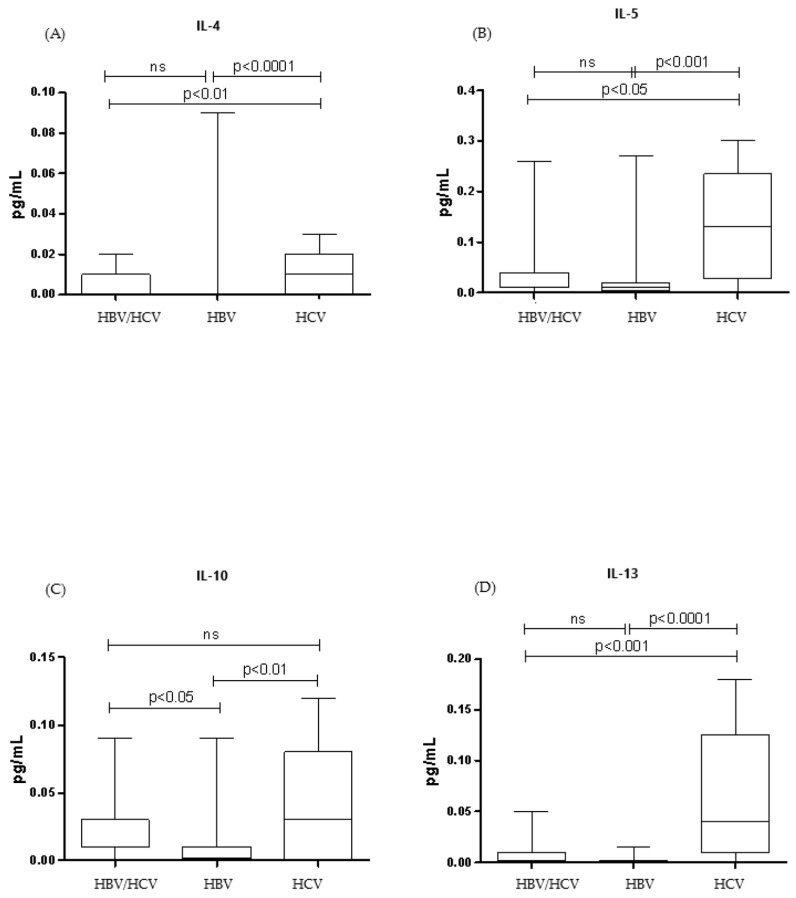
Th2 cytokine profile (anti-inflammatory). Comparison of IL-4 (**A**), IL-5 (**B**), IL-10 (**C**), and IL-13 (**D**) cytokine levels between co-infected (HBV/HCV), monoinfected (HBV), and monoinfected (HCV) groups. Data are expressed as median (25th–75th percentile). Kruskal–Wallis test.

**Figure 3 tropicalmed-10-00166-f003:**
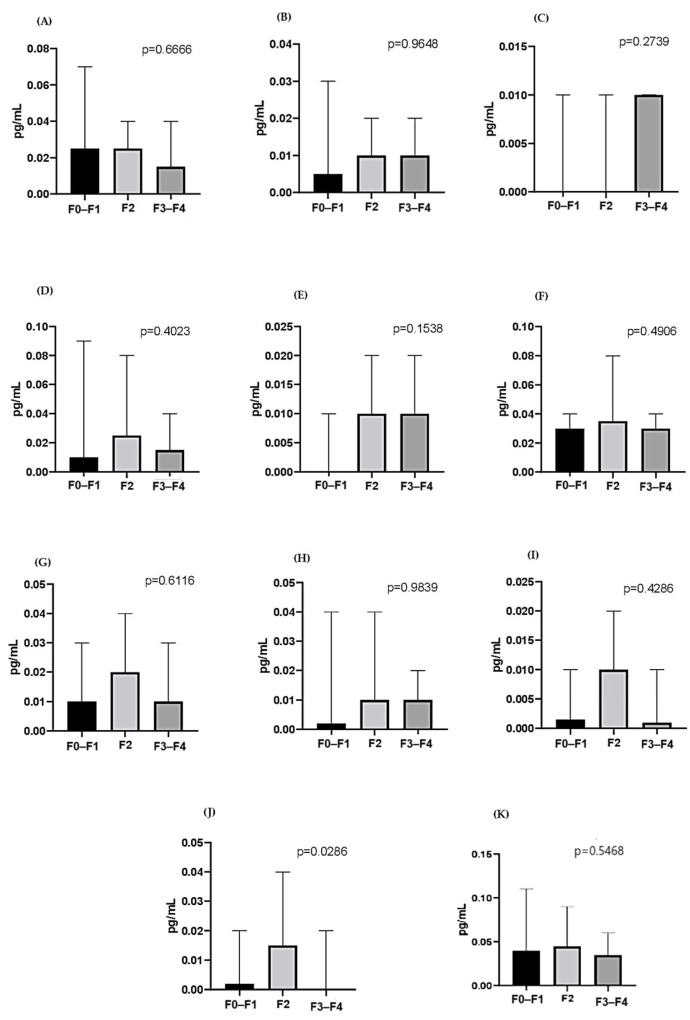
Comparison of cytokine levels in fibrosis scores F0–F1, F2, and F3–F4, according to the METAVIR classification, in the co-infected group. IFN-**γ** (**A**), IL-2 (**B**), IL-4 (**C**), IL-5 (**D**), IL-6 (**E**), IL-8 (**F**), IL-10 (**G**), IL-12 (**H**), IL-13 (**I**), IL-17 (**J**), and TNF-α (**K**). Data are expressed as median (25th–75th percentile). Comparisons between groups were performed using the Kruskal–Wallis test.

**Figure 4 tropicalmed-10-00166-f004:**
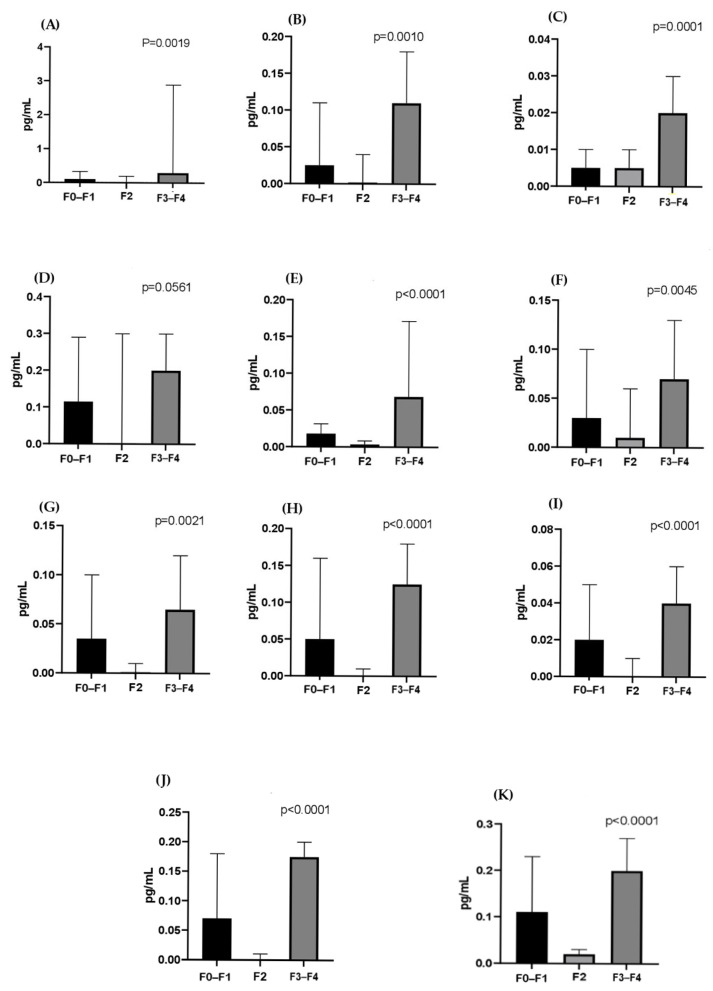
Comparison of cytokine levels in fibrosis scores F0–F1, F2, and F3–F4, according to the METAVIR classification, in the monoinfected (HCV) group. IFN-γ (**A**), IL-2 (**B**), IL-4 (**C**), IL-5 (**D**), IL-6 (**E**), IL-8 (**F**), IL-10 (**G**), IL-12 (**H**), IL-13 (**I**), IL-17 (**J**), and TNF-α (**K**). Data are expressed as median (25th–75th percentile). Comparisons between groups were performed using the Kruskal–Wallis test.

**Table 1 tropicalmed-10-00166-t001:** Demographic characteristics and laboratory data for HCV, HBV, and HBV/HCV.

	HBV/HCV (*n* = 34)	HBV (*n* = 22)	HCV (*n* = 22)	*p*
Age *	47.3 (±9.9)	42.1 (±9.4)	59.2 (±11.4)	<0.001
Sex M **	17 (50%)	17 (77.3%)	15 (68.2%)	0.098
Fibrosis (%) **				
F0/F1	16 (47.1)		6 (27.3)	
F2	12 (35.3)		6 (27.3)	0.050
F3/F4	6 (17.6)		10 (45.4)	
Viral genotype HCV (%) ***				
1	22 (64.7)		20 (91)	0.278
3	6 (17.6)		2 (9)	
Viral charge (IQR) ****				
DNA HBV	1.4 (0.22)	2.1 (0.77)		0.001
RNA HCV	5.5 (0.45)		6.1 (0.57)	0.005
Liver enzymes (IQR) *****				
AST (U/mL)	65.5 (30.5)	31 (9.5)	54.5 (30)	<0.001
ALT (U/mL)	65.0 (40.5)	30.0 (8.8)	79.5 (28.5)	<0.001

IQR, interquartile range. Data are expressed as median and average (IQR). Average * ANOVA ** Chi-Square test *** Fisher’s exact test **** Mann–Whitney test and Kruskal–Wallis test *****. *p* value < 0.05.

## Data Availability

The original contributions presented in this study are included in the article. Further inquiries can be directed to the corresponding authors.

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
