# Peer review of "Immunological and Virological Responses in Patients with Monoinfection and Coinfection with Hepatitis B and C Viruses in the Brazilian Amazon"

_tropicalmed, 2025, doi:10.3390/tropicalmed10060166_

Round 1
Reviewer 1 Report
Comments and Suggestions for Authors
Article “Immunological and virological response in patients with monoinfection and coinfection with hepatitis B and C viruses in the Brazilian Amazon” is devoted to study the immune response in Brazilian Amazon patients monoinfected with HCV and HBV and coinfected with HCV and HBV. In particular, we studied the levels of various cytokines, virus replication, as well as the state of the liver of patients, since these viruses predominantly affect this organ. The article is done at a good experimental level. All necessary experiments are given, illustrative material is given in good quality, discussion of the results is exhaustive. The shortcomings include a very small sample of patients, which is insufficient to obtain convincing results. Some necessary data, such as fibrosis in HBV monoinfected patients, are also missing, but the authors mention this themselves in the discussion of the results, and apparently do not have the technical ability to complete the study. It should be noted that these shortcomings cannot be addressed as the study is retrospective and was completed in 2019. Thus, the article can be published in its current form.

Reviewer 2 Report
Comments and Suggestions for Authors
- This study mentions the exclusion of patients with certain comorbidities, such as diabetes and hypertension. However, it would be useful to examine how other possible confounding factors, such as alcohol consumption or genetic factors, could affect the results. This should be addressed in the discussion.
- In the results section, fibrosis data were not reported for the hepatitis B mono-infected group. It would be useful to provide a discussion of the reasons for the absence of these data and their possible implications for the overall analysis.
- In certain sections of the manuscript, the terminology used to describe the virological response (such as "viral replication" versus "viral load") could be made more consistent to prevent any potential confusion.
- Discussion of proinflammatory cytokine profiles, particularly the dominance of IFN-γ and TNF-α in HCV mono-infection, is valuable. However, further elucidation of the mechanisms behind the reduction in cytokine levels in HBV mono-infection could provide further insight into the interpretation of the data.
Author Response
Por favor, veja o anexo

Reviewer 3 Report
Comments and Suggestions for Authors
This original study by Silva et al on "Immunological and virological response i patients with monoinfection and coinfection with HBV and HCV in the Brazilian Amazon", covers very interesting research and is generally well written. The article is easy to read, but has some minor flaws that need to be addressed by the authors:
Major points:
1- The laboratory tests should be better organized (lines 104-121). I recommend to start wit Biochemistry -> Serology -> PCR -> then HCV genotyping.
2- Were the included patients tested for anti-HBVc antibodies? What if they were PCR and HBsAg negative and were previously infected by HBV? This should be addressed in the article.
3- Figures 1 and 2: better to place the cytokine name under each histogram as it would facilitate the task for the reader.
4- Lines 198-204: The results are not correctly written or presented on figures 3 and 4. Please revisit all the p-values for the HCV monoinfected group.
5- Figures 3 and 4: There is an extra histogram (H) without the mention in the legend for both figures. Is it IL-12? Please correct accordingly.
6- Lines 218-220: The sentence about males is based on assumptions and should be remove. Or authors should provide more references from credible sources (in Brazil or in the world).
7- Line 225: This is not correct! HCV has more likelihood of becoming chronic in individuals. Please correct and add a reference to this claim.
8- Were HBV patients screened for hepatitis D coinfection with hepatitis B. There is a high prevalence in Northern parts of Brazil 8.5% according to a published study in 2018
Lago BV, Mello FCA, Barros TM, Mello VM, Villar LM, Lewis-Ximenez LL, Pardini MIMC, Lampe E; Brazilian Hepatitis B Research Group. Hepatitis D infection in Brazil: Prevalence and geographical distribution of anti-Delta antibody. J Med Virol. 2018 Aug;90(8):1358-1363. doi: 10.1002/jmv.25196. Epub 2018 May 1. PMID: 29663457
If this has not been done by the authors, it is advisable to test for hepatitis D coinfection, as it a bias in this study. Or add it to the limitations of the study.
Minor points:
1- Abstract line 26: ELISA in capital letters, but first should be fully spelled out.
2- Line 87: same comment for ELISA
3- Line 101: please add that the patient consent was obtained
4- line 115: Read is not a suitable word better used "determined" or analyzed by (spectrometry?)
5- Line 127: Please remove the dash from measurements
6- Line 140: The collected data was INSTEAD of The information obtained
7- line 146: The software was usd for what? for Graph generation?
8- Line 231: replace IP with IL
9- Line 279: Please add that ALT is more liver specific that AST and discuss accordingly
10- Line 295: Please remove the first "system"
11- Line 317-320: what are the implications of the results? what does the authors recommend?
Comments on the Quality of English LanguageThe article need some grammatical attention and better be read by a native english speaker.
Round 2
Reviewer 3 Report
Comments and Suggestions for Authors
The authors have addressed all the points that I raised and made all the necessary modifications. I have no further questions.
Comments on the Quality of English LanguageThe article still requires some grammatical attention. Better if it is reviewed by a native speaker.